# Parameter-Efficient Fine-Tuning of LLMs with Mixture of Space Experts

## Abstract

Large language models (LLMs) have achieved remarkable progress, with Parameter-Efficient Fine-Tuning (PEFT) emerging as a key technique for downstream task adaptation. However, existing PEFT methods mainly operate in Euclidean space, fundamentally limiting their capacity to capture complex geometric structures inherent in language data. While alternative geometric spaces, such as hyperbolic geometries for hierarchical data and spherical manifolds for circular patterns, offer theoretical advantages, constraining representations to single manifold types fundamentally limits expressiveness, even with learnable curvature parameters. To address this, we propose **MoS** (Mixture of Space), a unified framework that leverages multiple geometric spaces simultaneously to learn richer, curvature-aware representations. Building on this scheme, we develop **MoSELoRA**, which extends Low-Rank Adaptation (LoRA) with heterogeneous geometric experts, enabling models to dynamically select or combine appropriate geometric spaces based on input context. Besides, to address the computational overhead of frequent manifold switching, we develop a lightweight routing mechanism. Moreover, we provide empirical insights into how curvature optimization impacts training stability and model performance. Our experiments across diverse benchmarks demonstrate that MoSELoRA consistently outperforms strong baselines, achieving up to 5.6% improvement on MATH500 and 15.9% on MAWPS.

## 1 Introduction

Large language models (LLMs) have recently demonstrated impressive performance across a wide range of applications, including translation, comprehension, dialogue, and reasoning (Achiam et al., 2023; Jaech et al., 2024; Dubey et al., 2024; Team, 2024). With the aid of post-training techniques such as instruction tuning, they can be further adapted to diverse downstream tasks with notable gains in effectiveness (Hu et al., 2023; Han et al., 2024). Despite these advances, most existing approaches rely on a Euclidean assumption, modeling all embeddings in flat Euclidean space, which are often inadequate to capture the semantic diversity and contextual complexity of natural language (Bronstein et al., 2017; Park et al., 2024; He et al., 2025b).

Semantic structures in language often display geometric patterns: hierarchical relationships where broad and general concepts naturally encompass finer subcategories and distinct entities; circular patterns among synonymous expressions or co-referential terms; and complex multi-level dependencies that resist simple linear organization. These patterns are largely overlooked and constrained under Euclidean representations, leaving open the question of *how to effectively leverage such naturally occurring structures within embedding spaces to unlock richer representational capacity.*

Recently, growing attention has shifted toward non-Euclidean constant-curvature spaces as alternatives to Euclidean embeddings for improving model performance (Peng et al., 2021; Yang et al., 2024d; Pal et al., 2024; Loshchilov et al., 2024). From an embedding perspective, it has been observed that tokens associated with higher-level and more general semantics often occupy regions of lower norm, whereas tokens tied to more concrete and specific meanings are distributed in regions of higher norm (Yang et al., 2024c). Hyperbolic space, with its negative curvature and exponential growth capacity, offers an effective means of embedding complex hierarchical information in lower dimensions compared to Euclidean space. Building on these advantages, Yang et al. (2024c) explored combining LoRA with hyperbolic geometry, enabling efficient fine-tuning of pretrained

LLMs while reducing embedding distortion. Similarly, spherical manifolds have shown promise for capturing circular patterns and normalized representations, as demonstrated by Loshchilov et al. (2024) who reformulated Transformers as hyperspherical models (nGPT) by enforcing embeddings to lie on the unit sphere.

**Limitations of Existing Methods.** However, real-world language data exhibits complex, heterogeneous structural relationships that cannot be adequately captured by constraining representations to a single geometric space. For instance, a single sentence may contain both hierarchical semantic relationships (e.g., category-subcategory structures) and circular patterns (e.g., synonymous expressions or co-referential terms), requiring different geometric inductive biases simultaneously. Furthermore, existing non-Euclidean approaches face significant computational challenges. Prior work often employs exponential and logarithmic maps to transition between non-Euclidean and Euclidean spaces (Ganea et al., 2018; Yang et al., 2022), incurring substantial computational overhead at each model layer. These repeated mappings are difficult to scale to models with larger parameter counts and greater depth, creating a practical barrier to widespread adoption.

To address these fundamental limitations, we propose a unified **Mixture of Space (MoS)** framework that integrates three types of constant-curvature spaces—Hyperbolic, Spherical, and Euclidean—enabling the simultaneous capture of diverse geometric structures within a single model. Rather than constraining all representations to a single geometric paradigm, our approach allows different tokens to reside in the geometric space most suited to their structural properties: hierarchical concepts in hyperbolic space, circular patterns in spherical space, and general relationships in Euclidean space.

Building upon this framework, we introduce **MoSELoRA**, which combines the MoS paradigm with Low-Rank Adaptation for efficient fine-tuning of large language models. MoSELoRA employs a lightweight token routing mechanism that dynamically assigns each token to its optimal geometric expert, avoiding the computational overhead of repeated space transformations while maintaining the representational benefits of multiple geometries. This design enables the model to adapt its geometric inductive biases on-the-fly, matching the heterogeneous structural requirements of real-world language data. Our contributions can be summarized as follows:

- We introduce a unified architecture that integrates three distinct constant-curvature spaces, and combine it with Mixture-of-Experts (MoE) and Low-Rank Adaptation (LoRA) to form a novel and efficient fine-tuning framework for LLMs.

- We design a lightweight token routing mechanism that efficiently directs tokens among multiple geometric spaces to overcome high-overhead space transformation.

- We provide an in-depth analysis of the training dynamics of space selection and routing strategies, along with optimizing geometric space integration during fine-tuning.

- We evaluate the proposed method on benchmarks including natural language understanding and mathematical reasoning, where it consistently outperforms several strong baselines.

## 2 RELATED WORK

### 2.1 MIXTURE OF LoRA EXPERTS

MoE introduces multiple expert networks and a gating network that selects experts based on different data characteristics(Jacobs et al., 1991). Specifically, the Sparsely-Gated Mixture-of-Experts mechanism(Shazeer et al., 2017) improves the capacity and computational efficiency of large-scale models by selecting sparse combinations of experts. Recently, with the LLMs PEFT techniques, MoE architectures have begun to be extended to corresponding PEFT methods(Mangrulkar et al., 2022; Gao et al., 2022; Zadouri et al., 2024), particularly LoRA-based MoE methods(Feng et al., 2024; Wu et al., 2024b;a). Mini-Ensemble LoRA (MELoRA)(Ren et al., 2024) partitions the original LoRA matrices into smaller, equivalent submatrices along the diagonal, which are then concatenated to form a structure akin to the MoE architecture, achieving a higher effective rank with fewer parameters, while simultaneously reducing computational complexity by a factor of $n^2$. To improve parameter efficiency, HydraLoRA(Tian et al., 2024) proposes an asymmetric structure for LoRA experts, where a common LoRA matrix $W_A$ is shared across all experts, while each expert has its own distinct LoRA matrix $W_{B_i}$ further extending the LoRA-based MoE architectures. HMoRA(Liao

et al., 2025) employs a hierarchical hybrid routing strategy, combining token-level and task-level routing mechanisms, and proposes a novel routing auxiliary loss function. This design not only enhances the task router's ability to distinguish tasks, but also more effectively captures fine-grained token information and broader task context, resulting in remarkable performance in multi-task learning. However, these approaches often struggle with the trade-off between parameter efficiency and model performance, as increasing the number of activated parameters can lead to higher computational costs and reduced efficiency. Our work extends LoRA experts to heterogeneous geometric space, compared to the original experts, offering stronger expressiveness and achieving a better balance between fewer activated parameters and model performance.

## 2.2 NON-EUCLIDEAN AND CURVATURE-AWARE MODELING

Recent research increasingly emphasized the importance of non-Euclidean geometry for representation learning, aiming to overcome the limitations of standard Euclidean embedding spaces (Shimizu et al., 2020; Peng et al., 2021; Pal et al., 2024; He et al., 2025c;b). Early attempts focused on constructing neural networks that operated fully in hyperbolic space. For example, Chen et al. (2022) formalized all operations as Lorentz transformations, thereby avoiding the reliance on tangent-space approximations. Beyond purely hyperbolic models, Skopek et al. (2020) introduced mixed-curvature variational autoencoders, whose latent spaces were composed of multiple constant-curvature manifolds, enabling generative models to benefit from diverse geometric structures simultaneously. More recently, Yang et al. (2024d) explored an efficient Transformer architecture in the Lorentz model of hyperbolic space, providing hyperbolic counterparts for essential modules such as positional encodings, layer normalization, and residual connections. Parallel to architectural advances, Yang et al. (2024c) investigated fine-tuning Euclidean LLMs directly in hyperbolic space, demonstrating improved downstream performance by leveraging the inherent hierarchical structure of token embeddings. Building on this line of work, He et al. (2025a) introduced Hyperbolic Large Language Models with a Mixture-of-Curvature Experts design, where each expert resided in a hyperbolic manifold of distinct curvature, allowing flexible encoding of input sequences and showcasing the scalability of curvature-aware modeling in large-scale pretraining.

## 3 PRELIMINARY

### 3.1 MIXTURE OF LORA EXPERTS

Mixture of LoRA Experts consists of a group of $N$ uniform experts $\{E_i\}_{i=1}^N$, where each features a parameter-efficient LoRA module to store updated parameters while fine-tuning. Each expert $E_i$ has the following forward process:

$$E_i = B_i A_i X, \tag{1}$$

where the tunable weight matrices $A_i \in \mathbb{R}^{r \times d_{\text{in}}}$, $B_i \in \mathbb{R}^{d_{\text{out}} \times r}$, and $r \ll \min\{d_{\text{in}}, d_{\text{out}}\}$ is the maximum rank attainable by the trainable matrix, with matrix $A_i$ randomly initialized and matrix $B_i$ set to all zero. The Mixture of LoRA Experts forward process can be formulated as follows:

$$O = WX + \sum_{i=1}^N G(X)_i E_i = WX + \sum_{i=1}^N G(X)_i B_i A_i X, \tag{2}$$

where $W$ is the frozen pretrained weight of the feed-forward neural network (FFN) block, $G(X)$ is the token router in the MoE module which routes each token into several distinct experts (eg., top-$K$ experts for sparse MoE (Fedus et al., 2022)) among all $N$ experts.

### 3.2 LORENTZ MODEL OF HYPERBOLIC SPACE

**Lorentz Model.** The Lorentz model, also called the hyperboloid model, provides one of the isometric realizations of the hyperbolic space as a Riemannian manifold. Formally, an $n$-dimensional Lorentz model with constant negative curvature $\kappa < 0$ is defined as

$$\mathbb{L}^{n,\kappa} = \left\{ \mathbf{x} \in \mathbb{R}^{n+1} \mid \langle \mathbf{x}, \mathbf{x} \rangle_{\mathcal{L}} = 1/\kappa, \ x_t > 0 \right\},$$

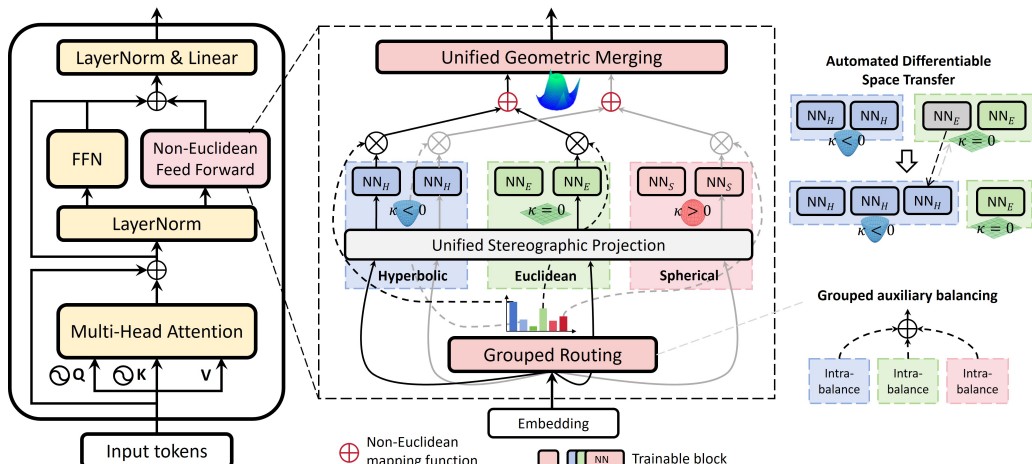

Figure 1: The MoSELoRA architecture contains heterogeneous geometric experts unified in our MoS scheme, with various curvatures. Three geometric expert groups are embedded into the FFN layer of LLMs and labeled in three different colored blocks. The grouped auxiliary balancing enforces balanced routing within each space group while allowing free inter-space transitions, and remains fully reversible and differentiable.

where $\mathbf{x} = [x_t; \mathbf{x}_s]^\top$ with $x_t \in \mathbb{R}$ and $\mathbf{x}_s \in \mathbb{R}^n$, and the Lorentzian inner product is given by

$$\langle \mathbf{x}, \mathbf{y} \rangle_{\mathcal{L}} = -x_t y_t + \mathbf{x}_s^\top \mathbf{y}_s = \mathbf{x}^\top \mathrm{diag}(-1, 1, \ldots, 1)\, \mathbf{y}.$$

Geometrically, $\mathbb{L}^{n,\kappa}$ corresponds to the upper sheet of a two-sheeted hyperboloid embedded in $(n + 1)$-dimensional Minkowski space, with the distinguished coordinate $x_t$ representing the time-like axis and the remaining $n$ coordinates forming the space-like axes. This construction not only aligns with the terminology of special relativity (Resnick, 1991) but also ensures numerical stability in optimization tasks.

**Tangent Space and Maps.** For each point $\mathbf{x} \in \mathbb{L}^{n,\kappa}$, the tangent space $\mathcal{T}_{\mathbf{x}}\mathbb{L}^{n,\kappa}$ is defined as the Lorentz-orthogonal complement of $\mathbf{x}$ and constitutes a smooth Euclidean subspace of $\mathbb{R}^{n+1}$. This tangent space provides a local linear approximation of the curved hyperbolic manifold, which is fundamental for optimization and representation learning. The transition between the manifold and its tangent space is realized through the exponential and logarithmic maps.

The exponential map $\exp_{\mathbf{x}}^{\kappa} : \mathcal{T}_{\mathbf{x}}\mathbb{L}^{n,\kappa} \to \mathbb{L}^{n,\kappa}$ takes a tangent vector $\mathbf{u} \in \mathcal{T}_{\mathbf{x}}\mathbb{L}^{n,\kappa}$ and projects it onto the manifold along the geodesic starting at $\mathbf{x}$:

$$\exp_{\mathbf{x}}^{\kappa}(\mathbf{u}) = \cosh\left(\sqrt{|\kappa|}\|\mathbf{u}\|_{\mathcal{L}}\right)\mathbf{x} + \frac{\sinh\left(\sqrt{|\kappa|}\|\mathbf{u}\|_{\mathcal{L}}\right)}{\sqrt{|\kappa|}\|\mathbf{u}\|_{\mathcal{L}}}\,\mathbf{u}.$$

Conversely, the logarithmic map $\log_{\mathbf{x}}^{\kappa} : \mathbb{L}^{n,\kappa} \to \mathcal{T}_{\mathbf{x}}\mathbb{L}^{n,\kappa}$ takes a point $\mathbf{y} \in \mathbb{L}^{n,\kappa}$ and returns the unique tangent vector at $\mathbf{x}$ that corresponds to the geodesic connecting $\mathbf{x}$ and $\mathbf{y}$:

$$\log_{\mathbf{x}}^{\kappa}(\mathbf{y}) = \frac{\cosh^{-1}\left(\kappa\langle \mathbf{x}, \mathbf{y} \rangle_{\mathcal{L}}\right)}{\sinh\left(\cosh^{-1}(\kappa\langle \mathbf{x}, \mathbf{y} \rangle_{\mathcal{L}})\right)}\left(\mathbf{y} - \kappa\langle \mathbf{x}, \mathbf{y} \rangle_{\mathcal{L}}\mathbf{x}\right).$$

These two maps establish a rigorous correspondence between the locally Euclidean tangent space and the globally curved hyperbolic manifold, ensuring that vectors can be transferred consistently between the two domains.

## 4 OUR MOS AND MOSELORA FRAMEWORK

In this section, we propose our curvature-aware tuning scheme that adapts the model to the implicit curvature of semantic subspaces while autonomously exploring and transitioning between different constant curvatures within the same subspace, and also across distinct subspaces.

We start by constructing a unified framework that accommodates embeddings from spaces with different curvatures, including Euclidean, Hyperbolic, and Spherical geometries. Such a formulation enables representations to be jointly learned across multiple manifolds, thereby capturing complementary structural information and offering diverse perspectives of the embedding knowledge. By integrating these curved spaces within a single scheme, the model can flexibly adapt to heterogeneous relational patterns while maintaining consistency across geometries. Moreover, the unified design is implemented in a computationally efficient manner, ensuring that the transformation between different manifolds can be seamlessly incorporated into the forward pass of modern neural architectures. Afterwards, we will outline our curvature-aware tuning scheme with a mixture of geometry-integrated experts.

**Mixture of Space scheme.** As previously introduced, the standard Mixture of LoRA Experts scheme comprises a group of isomorphic experts, which are intended to exploit the combinatorial richness offered by the large number of possible substructure configurations. However, the backbone of these combinations still resides in a flat (Euclidean) space, which is characterized by polynomial volume growth with respect to radius, and therefore overlooks the intrinsic geometry of the embedding space.

Therefore, we proposed to integrate non-Euclidean geometry into each expert to equip each layer with curvature-aware representation capabilities, capturing underlying features in different depths. Specifically, we explore three distinct constant curvature spaces characterized by positive, negative, and zero curvature, and common space models for these three curved spaces are Spherical $\mathbb{S}$, Hyperbolic $\mathbb{H}$, and Euclidean $\mathbb{E}$ (flat) space. For a $n$-dimensional hyperbolic space $\mathbb{H}_\kappa^n$ with constant negative curvature $\kappa$, each point $\mathbf{x} \in \mathbb{R}^{n+1}$ in $\mathbb{H}_\kappa^n$ should satisfy the following definition:

$$\mathbb{H}_\kappa^n := \{\mathbf{x} \in \mathbb{R}^{n+1} | \langle \mathbf{x}, \mathbf{x}\rangle_L = 1/\kappa, \kappa < 0\}, \quad \langle \mathbf{x}, \mathbf{x}\rangle_L = -x_1^2 + \sum_{i=2}^{n+1} x_i^2, \tag{3}$$

where $\mathbb{H}_\kappa^n$ is defined by the Lorentz inner product $\langle \mathbf{x}, \mathbf{x}\rangle_L$, and $\mathbf{x} = [x_1, x_2, \ldots, x_{n+1}] = [x_{\text{time}}, \mathbf{x}_{\text{space}}]$ denotes an arbitrary point with time-like component $x_1$ and space-like component $[x_2, \ldots, x_{n+1}]$. In the Lorentz model of hyperbolic space, the volume $V_\kappa$ grows exponentially with its radius $r$ as $V_\kappa(r) \asymp \exp\left((n-1)\sqrt{-\kappa}\, r\right)$. Thus, the larger the magnitude of $K$, the greater the curvature and the faster the volume expansion, allowing the space to accommodate more hierarchical structures with higher representational capacity. For spherical spaces with curvature $\kappa > 0$, each point $\mathbf{x} \in \mathbb{R}^{n+1}$ should satisfy:

$$\mathbb{S}_\kappa^n := \{\mathbf{x} \in \mathbb{R}^{n+1} | \langle \mathbf{x}, \mathbf{x}\rangle_2 = 1/\kappa, \kappa > 0\}, \quad \langle \mathbf{x}, \mathbf{x}\rangle_2 = \sum_{i=1}^{n+1} x_i^2, \tag{4}$$

where $\langle \cdot, \cdot\rangle_2$ is the Euclidean inner product. Unlike hyperbolic spaces which suits for representing complex hierarchical structures, volume in spherical spaces grows sub-exponentially with radius, hence they are especially suitable for modeling cyclic, periodic, or bounded structures, where global capacity is limited but dense local clustering and angular relationships (e.g., directions, orientations) are crucial. In natural language, for instance, temporal expressions such as the days of the week (e.g., Monday, Tuesday, ..., Sunday) or months in a year (e.g., January, February, ..., December) exhibit intrinsic cyclic structures. These patterns are not only semantic but manifest as circular topologies in the embedding space (Engels et al., 2024). Combined with the default Euclidean space, we propose a unified paradigm suitable for LLMs parameter-efficient fine-tuning without relying on computationally expensive and GPU-unfriendly operations such as exponential and logarithmic maps, which are standard space transition methods used by previous methods (Chen et al., 2021; Bdeir et al., 2023; Yang et al., 2024b). Instead, we adopt a unified Lorentz model framework, which, for the first time, enables consistent and efficient bidirectional stereographic transformations across all subspaces within a single cohesive architecture.

First, each token $\boldsymbol{x}_i$ (in the Euclidean space) in the token embedding sequence from the previous layer will be transmitted into a projected space through stereographic conformal inverse projection $\rho_\kappa^{-1}(\cdot)$ in equation 5 to $y_i = (\xi_i; \boldsymbol{s}_i^T)^T \in \mathbb{R}^{n+1}, 1 \le i \le Q$, where $\kappa \in \mathbb{R}$ is the curvature of the projected embedding space.

$$\rho_\kappa^{-1}(\boldsymbol{x}_i) = \left(\frac{1}{\sqrt{|\kappa|}} \frac{1 - \kappa\|\boldsymbol{x}_i\|_2^2}{1 + \kappa\|\boldsymbol{x}_i\|_2^2}, \frac{2\boldsymbol{x}_i}{1 + \kappa\|\boldsymbol{x}_i\|_2^2}\right)^T = (\xi_i; \boldsymbol{s}_i^T)^T \tag{5}$$

Then, the space-like component $s_i \in \mathbb{R}^n$ of the projected embedding $(\xi_i; s_i^T)^T \in \mathbb{R}^{n+1}$ will be treated as the unit routing token for our mixture of unified geometric experts (in each layer's FFN block). Our unified mixture of space experts is defined as follows:

$$y_i' = \text{GE}(W, \boldsymbol{x}_i) = \begin{bmatrix} \sqrt{\|\phi(W\boldsymbol{s}_i)\|^2 \cdot \mathbf{sgn}(-\kappa) + \varphi(\kappa)} \\ \phi(W\boldsymbol{s}_i) \end{bmatrix} = \begin{bmatrix} \xi_i' \\ \boldsymbol{s}_i' \end{bmatrix}, \quad (6)$$

$$\mathbf{sgn}(\kappa) := \begin{cases} -1 & \text{if } \kappa < 0, \\ 1 & \text{if } \kappa \geq 0, \end{cases} \quad \varphi(\kappa) := \begin{cases} 1/|\kappa| & \text{if } \kappa \neq 0, \\ 0 & \text{if } \kappa = 0. \end{cases} \quad (7)$$

where $\text{GE}(W, \boldsymbol{x}_i)$ denotes the unified neural component of different geometric experts with different signs of the curvature $\kappa$ and module weights $W$. We restrict inputs to the space-like component, as Lorentz geometry in $\mathbb{R}^{n+1}$ provides only $n$ degrees of freedom, yielding better numerical stability.

**Mixture of Space Experts.** The output tokens from the top-$K$ geometric experts, each operating on distinct curved spaces and capturing various hierarchical or concurrent relations, are projected back into Euclidean space, where these features are preserved and merged together. Our **MoSELoRA** is illustrated in Figure 1, and the non-Euclidean feed-forward output is formulated as follows:

$$\boldsymbol{o}_i = \sum_{i=1}^N \sum_{j=1}^Q G(\boldsymbol{x}_j)_i \, \rho_{\kappa_i}\left(\left(\xi_j; \boldsymbol{s}_j'^T\right)^T\right), \quad \rho_{\kappa_i}((\xi_j; \boldsymbol{s}_j'^T)^T) = \frac{\boldsymbol{s}_j'}{1 + \sqrt{|\kappa_i|}\xi_j'} \quad (8)$$

where $\rho_{\kappa_i}((\cdot; \cdot))$ is the stereographic projection to map points (token embeddings) from the curved space back to the Euclidean space, $\kappa_i$ is the curvature of the $i$-th geometric expert, $G(x_j)_i$ is the routing value for token $x_j$ and $i$-th expert, and we utilize the auxiliary loss to balance top-$k$ routing across heterogeneous geometric experts. The curvature $\kappa$ of each expert is a learnable parameter, so that the model can independently adjust the sub-space of each expert to approach the optimal geometry. It is worth noting that the stereographic mapping and the associated routing strategy used here require neither complex exponential/logarithmic space operations nor prior knowledge of the latent embedding space, thereby yielding higher efficiency in both training and inference. The analysis and proof of gradient boundedness for our framework can be found in the Appendix C. We also give a Considering that curvature, as a geometric parameter, differs fundamentally from other capacity-related parameters, and in order to avoid being trapped in local sub-optima, we assign curvature an independent optimizer and optimization path to encourage further exploring the geometric curvature of the latent space while reducing the risk of overfitting to local data patterns. Formally, we have

$$\Theta = \{\kappa^{(1)}, \ldots, \kappa^{(K)}, \; \theta^{(1)}, \ldots, \theta^{(M)}\}, \quad (9)$$

$$w \; \leftarrow \; w - \sum_{j=1}^N \eta_\kappa^{(j)} \, \mathbf{1}_{w \in \kappa^{(j)}} \cdot g_\kappa^{(j)}(w) - \sum_{m=1}^M \eta_\theta^{(m)} \, \mathbf{1}_{w \in \theta^{(m)}} \cdot g_\theta^{(m)}(w), \quad (10)$$

where $\Theta$ consists of curvature parameters $\kappa^{(j)}$ and capacity-related parameters $\theta^{(m)}$. $\eta_\kappa^{(j)}$ and $\eta_\theta^{(m)}$ denote their respective learning rates, $g_\kappa^{(j)}(\cdot)$ and $g_\theta^{(m)}(\cdot)$ are the corresponding gradients, and $\mathbf{1}_{(\cdot)}$ is an indicator function selecting the parameter group.

# 5 EXPERIMENTS

## 5.1 EXPERIMENT SETTING

**Dataset and Benchmarks.** To explore the utility of our methods, we evaluate our approach on both natural language understanding and mathematical reasoning datasets. For NLP tasks, we adopt the Microsoft Research Paraphrase Corpus (MRPC) (Dolan & Brockett, 2005), which consists of sentence pairs labeled as paraphrases or not, and the Corpus of Linguistic Acceptability (CoLA) (Warstadt et al., 2019), which contains expert judgments on the grammatical acceptability of English sentences(Wang et al., 2018). For mathematical reasoning, we employ GSM8K (Cobbe et al., 2021), MAWPS (Koncel-Kedziorski et al., 2016), SVAMP (Patel et al., 2021), and AQuA(Ling et al., 2017), alongside the MATH500 by OpenAI (Lightman et al., 2023), a curated subset of the MATH benchmark (Hendrycks et al., 2021), comprising 500 challenging problems that span algebra, geometry, intermediate algebra, number theory, precalculus, and probability. The training set

Table 1: Performance comparison of different tuning methods across different natural language understanding and mathematical reasoning benchmarks. The Avg. column reports the average score, and rank ↑ indicates that a larger rank parameter is used compared to other methods.

| Methods | COLA | MRPC | GSM8k | MATH500 | MAWPS | SVAMP | AQuA | Avg. |
|---|---|---|---|---|---|---|---|---|
| Base | 32.41 | 69.16 | 33.13 | 0.0 | 43.85 | 54.33 | 31.10 | 37.71 |
| LoRA | 87.25 | 87.30 | 44.58 | 15.60 | 63.85 | 63.33 | 28.35 | 55.75 |
| DoRA | 87.15 | 87.19 | 42.53 | 14.20 | 62.12 | 62.33 | 30.71 | 55.18 |
| LoRA rank ↑ | 83.51 | 86.43 | 46.85 | 13.60 | 61.92 | 60.33 | **33.46** | 55.16 |
| AdaLoRA | 82.45 | 79.77 | **51.33** | 16.60 | 62.50 | 58.67 | 23.23 | 53.51 |
| MELoRA | 86.96 | 86.55 | 44.96 | 16.40 | 62.69 | **65.67** | 32.28 | 56.50 |
| HMoRA | 66.63 | 66.49 | 12.28 | 2.60 | 54.23 | 46.00 | 24.41 | 38.95 |
| HydraLoRA | 87.54 | 88.17 | 43.67 | 16.80 | 62.88 | 62.67 | 28.35 | 55.73 |
| **MoSELoRA (Ours)** | **87.63** | **88.23** | 47.23 | **17.80** | **75.96** | 64.67 | 30.71 | **58.89** |

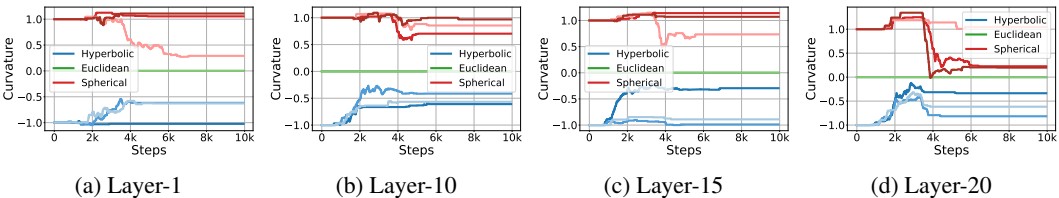

(a) Layer-1      (b) Layer-10      (c) Layer-15      (d) Layer-20

Figure 2: Curvature dynamics of each geometric expert of different layers during training.

is constructed by uniformly sampling from the mathematical datasets except MATH500, which is reserved entirely for evaluation, thereby enabling rigorous assessment of zero-shot and few-shot performance on unseen problems.

**Base Model and Baselines.** We include Qwen2-1.5B-Base (Yang et al., 2024a) as our base model. For parameter-efficient fine-tuning methods, we evaluate our methods with LoRA (Hu et al., 2022), AdaLoRA (Zhang et al., 2023), DoRA (Liu et al., 2024), MELoRA (Ren et al., 2024), HMoRA (Liao et al., 2025), and HydraLoRA (Tian et al., 2024). These methods share the same training settings as ours, and several recent approaches also incorporate architectures related to the mixture of experts design. For other methods such as MoLA and AlphaLoRA, we consider them orthogonal to our approach. In this work, we instead adopt an asymmetric architecture similar to Tian et al. (2024), aiming to minimize the number of activated parameters while verifying the effectiveness of our proposed scheme.

**Training settings and Evaluation metrics.** All models are trained for three epochs on the same dataset using NVIDIA A100 (80G) and H800 (80G) GPUs. For MoE-based methods, we adopt eight experts with top-4 routing and apply the standard auxiliary load-balancing loss with the coefficient set to 0.01. To ensure fairness, we control the number of activated parameters across methods. For non-MoE baselines such as LoRA, we additionally match settings by adjusting the adaptation rank.

## 5.2 BASELINE COMPARISON

We compare our methods with other baselines using 8 distinct experts with top-4 routing with auxiliary load balance loss. In our method, three geometric experts are allocated for each non-Euclidean space group and two for the Euclidean group, with curvatures initialized to –1 for negative curvature and 1 for positive curvature spaces. To keep the number of activated parameters comparable across methods, we vary the LoRA rank from 8 to 64 (0.45%–3.53% of activated parameters), while fixing the rank at 8 for all other baselines. For MoE-related approaches such as HMoRA and HydraLoRA, the number of experts is set to 8. Due to their fully activated computation, HMoRA and HydraLoRA yield 2.26% and 2.54% activated parameters, respectively, whereas our method requires only 1.31%.

**Main Results.** As shown in Table 1, our **MoSELoRA** achieves higher average performance than existing state-of-the-art methods across tasks in both natural language understanding and mathematical reasoning benchmarks. In particular, it obtains the best results on most mathematical reasoning benchmarks, including the challenging MATH500 dataset. Since our training data is uniformly

Table 2: Performance of different optimizer choices across different MATH and general reasoning benchmarks. UNI and SEP denote using a unified optimizer and separated optimizers, respectively.

| Methods | COLA | MRPC | GSM8k | MATH500 | MAWPS | SVAMP | AQuA | Avg. |
|---|---|---|---|---|---|---|---|---|
| MoSELoRA UNI | 87.15 | 87.19 | 46.85 | 17.80 | 74.62 | 64.33 | 29.13 | 58.15 |
| MoSELoRA SEP | **87.63** | **88.23** | **47.23** | **17.80** | **75.96** | **64.67** | **30.71** | **58.89** |

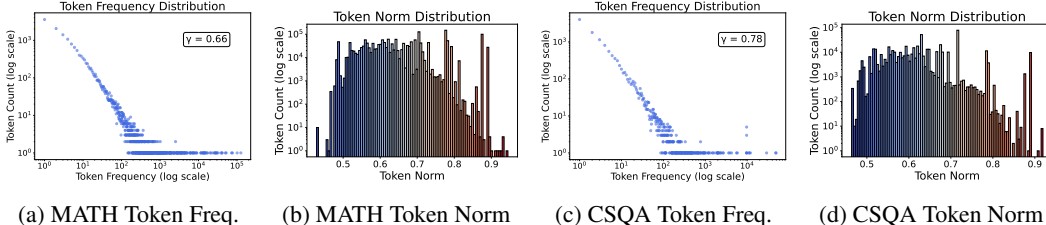

(a) MATH Token Freq.  (b) MATH Token Norm  (c) CSQA Token Freq.  (d) CSQA Token Norm

Figure 3: Token norm distribution and token frequency distribution of math reasoning and commonsenseQA datasets.

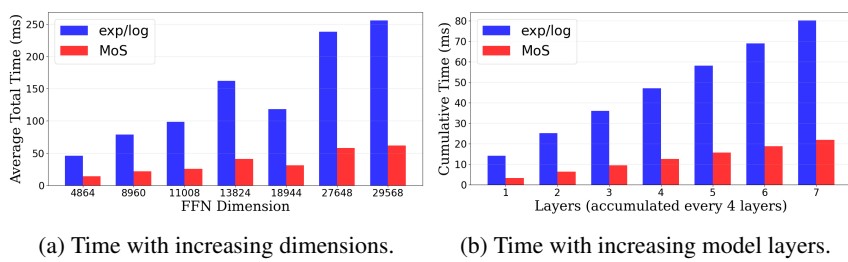

(a) Time with increasing dimensions.  (b) Time with increasing model layers.

Figure 4: Conversion time comparison between exp/log mapping and our MoS framework.

sampled from other mathematical reasoning datasets, these results further demonstrate the superior generalization ability of our **MoSELoRA** when tackling previously unseen reasoning problems. We observe that our method achieves particularly notable improvements on mathematical reasoning benchmarks compared to existing baselines. We attribute this to the introduction of the mixed-curvature framework, where hyperbolic space is especially well suited for representing hierarchical numerical and logical structures (Yang et al., 2024b), while spherical space can better capture cyclic properties such as equivalence relations commonly present in mathematics. Consequently, unlike Euclidean baselines that rely solely on a flat embedding space, our approach provides clear advantages in handling large-number computations and symbolic operations.

### 5.3 TUNING DYNAMICS

**Curvature training dynamics of distinct geometric experts.** In our MoSELoRA framework, the curvature parameter $\kappa$ is distinct from other capacity parameters $\theta$, serving as a key geometric property that simultaneously characterizes both the input latent space and the model's embedding space. As shown in Figure 2, we track the evolution of embedding spaces for geometric experts across model layers. Distinct colors denote different space types, with varying intensities representing experts within the same type. To provide stable Euclidean embeddings, two Euclidean experts with fixed curvature were included in all experiments. During training, the geometric experts progressively selected the curvature spaces that minimized embedding loss, and after approximately 6k–8k steps these selections stabilized, suggesting that the model had automatically identified the optimal mixture of embedding spaces under the given training configuration. We also observed interesting dynamic behaviors, such as recurrent switching across curvature spaces and transitions between geometric spaces. For instance, in Fig. 2d, one spherical expert in the 20th layer temporarily shifted into the Euclidean space around step 4k, but subsequently returned to the spherical space and ultimately converged to a positive curvature near 0.2. These dynamics indicate that the model can adaptively adjust combinations of geometric experts across spaces, highlighting the flexibility and stability

Table 3: Performance comparison of different mixture recipes of geometric experts across different mathematical and general reasoning benchmarks.

| Methods | Space | COLA | MRPC | GSM8k | MATH500 | MAWPS | SVAMP | AQuA | Avg. |
|---|---|---|---|---|---|---|---|---|---|
| **MoSELoRA-S** | Spherical | 86.86 | 87.77 | 44.50 | 17.60 | 72.69 | 64.33 | 28.35 | 57.44 |
| **MoSELoRA-H** | Hyperbolic | 85.81 | **89.33** | 45.34 | **17.80** | 73.46 | 63.67 | **34.25** | 58.52 |
| **MoSELoRA-E** | Euclidean | 87.15 | 87.19 | 42.23 | 16.20 | 70.77 | 61.67 | 31.89 | 56.73 |
| **MoSELoRA (Ours)** | Mixture | **87.63** | 88.23 | **47.23** | **17.80** | **75.96** | **64.67** | 30.71 | **58.89** |

of our **MoS** architecture as a foundation for **MoSELoRA**. Furthermore, layer-wise trends reveals that lower layers, particularly the first layer, exhibit more stable but slower convergence compared to higher layers. This aligns with observations in MoE-related literature (Dai et al., 2024; Muennighoff et al., 2024): early layers primarily capture task-agnostic token-level information, leading to smoother geometry selection, while higher layers encode more task-specific information, resulting in more pronounced cross-space transitions and dynamic curvature patterns, thereby underscoring the necessity of incorporating non-Euclidean geometry for downstream tasks.

**Optimizer for geometric experts.** We observed that when using a unified optimizer, the curvature $\kappa$ fail to adjust alongside other parameters, resulting in nearly static values and relatively poor task performance (in Table 2). Therefore, we assign independent optimizers and learning rates to $\kappa$ of all geometric experts, enabling the model to adjust latent geometric spaces more flexibly without being constrained by the optimization trajectory of other capacity-related parameters. Empirically, this setting leads to improved performance on downstream tasks.

**Token frequency distribution.** To further quantify and analyze the underlying geometric relationships from the perspective of token-level statistics, in Fig. 3, we examine both the token frequency distributions and the relationship between embedding norms and frequency of tokens produced by the tokenizers of Qwen2.5 models. We observe that the token frequency distribution exhibits a scale-free behavior, reflecting the latent hyperbolic characteristics of the underlying training corpus. Meanwhile, the token-norm distribution shows an exponential growth pattern on the right tail, whereas the left side deviates from such behavior. Prior work has partially observed similar phenomena, but their analysis focuses primarily on hyperbolic spaces, which are particularly well-suited for modeling power-law structures—yet less capable of capturing the broader range of non-Euclidean geometries that emerge across different contexts and domains.

**Geometric mapping efficiency.** In Figure 4, we compare the runtime on GPUs between our proposed lightweight routing and space-mapping method and the conventional approach based on exponential and logarithmic mappings. Figure 4a further illustrates how computation time scales with the FFN dimension across real models of different sizes, while Figure 4b shows the corresponding trend with respect to the number of model layers. The results show that our method achieves significant speedups up to $4\times$ over the standard exp–log scheme, and this relative acceleration remains consistent as the network depth increases and the embedding dimension grows.

## 5.4 ABLATION STUDY

**Different space-mix of experts.** To obtain a more fine-grained understanding of how different geometric expert mixture configuration affect performance, we conducted an ablation study by restricting **MoSELoRA** to use only a single type of expert. For example, in the hyperbolic-only setting, each expert is initialized with a negative curvature (e.g., –1), and during the training stage, curvatures are learnable so that the model can dynamically adjust the underlying embedding space for inputs. All other configurations follow our default setup, including the unified framework, lightweight routing strategy, top-4 out of 8 expert selection, and the auxiliary load-balancing loss. As shown in Table 3, on average, **MoSELoRA** with mixed-geometric experts consistently outperforms all single-space variants. Nevertheless, specific single-space configurations achieved strong results on certain datasets. For instance, the hyperbolic-only variant **MoSELoRA-H** achieves accuracies of 17.80% on MATH500, 34.25% on AQuA, and 89.33% on MRPC, matching or surpassing all competing methods, suggesting that hyperbolic embeddings are particularly well-suited for these benchmarks. The strong performance of hyperbolic experts on MATH500 further indicates their ability to generalize effectively in zero-shot settings to previously unseen reasoning problems. On the other hand, the Euclidean variant **MoSELoRA-E** outperforms others on CoLA (87.15%), while the spherical vari-

ant **MoSELoRA-S** achieves the best results on SVAMP (64.33%), highlighting that certain datasets benefit more from specific geometries. Together, these findings confirm the necessity of combining multiple spaces, as **MoSELoRA** effectively integrates the complementary strengths of different embedding geometries to achieve superior overall performance.

# 6 CONCLUSION

This paper first introduces MoS, a unified framework that integrates distinct geometric spaces and enables flexible transformations among three constant-curvature spaces: Hyperbolic, Euclidean, and Spherical space. Building on this framework, we further propose MoSELoRA, an efficient fine-tuning architecture for LLMs that combines mixture-of-space experts. This design allows LLMs to dynamically adjust the curvature of each expert's underlying space during fine-tuning and to flexibly reconfigure combinations of different spaces based on the input. In addition, to address the computational overhead of manifold-based exponential and logarithmic operations commonly adopted in existing non-Euclidean models, we develop a lightweight routing and space-mapping strategy that improves the efficiency of space transitions. Experimental results demonstrate that our approach consistently outperforms strong baselines and provides new insights into geometric representation learning. Nevertheless, further investigation is required to assess its applicability to industrial-scale settings, as well as to more recent reasoning and reinforcement learning frameworks.

# 7 ETHICS STATEMENT

This study follows the ICLR Code of Ethics. In this study, no human subjects or animal experimentation was involved. All datasets used, including Microsoft Research Paraphrase Corpus (MRPC) (Dolan & Brockett, 2005), Corpus of Linguistic Acceptability (CoLA) (Warstadt et al., 2019), GSM8K (Cobbe et al., 2021), MAWPS (Koncel-Kedziorski et al., 2016), SVAMP (Patel et al., 2021), AQuA (Ling et al., 2017) and MATH500 (Lightman et al., 2023), were sourced in compliance with relevant usage guidelines, ensuring no violation of privacy. We have taken care to avoid any biases or discriminatory outcomes in our research process. No personally identifiable information was used, and no experiments were conducted that could raise privacy or security concerns. We are dedicated to upholding transparency and integrity throughout the course of this research.

# 8 REPRODUCIBILITY STATEMENT

We have made every effort to ensure that the results presented in this paper are reproducible. All code and datasets have been made publicly available in an anonymous repository to facilitate replication and verification. The experimental setup, including training steps, model configurations, and hardware details, is described in detail in the paper. We have also provided a full description of PEFT of LLMs with mixture of space experts, to assist others in reproducing our experiments. Additionally, all language understanding and mathematical reasoning datasets, such as MRPC(Dolan & Brockett, 2005), CoLA(Warstadt et al., 2019), GSM8K (Cobbe et al., 2021), MAWPS (Koncel-Kedziorski et al., 2016), SVAMP (Patel et al., 2021), AQuA (Ling et al., 2017) and MATH500 (Lightman et al., 2023), are publicly available, ensuring consistent and reproducible evaluation results. We believe these measures will enable other researchers to reproduce our work and further advance the field.

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

## A  THE USE OF LARGE LANGUAGE MODELS (LLMS)

During the writing of this paper, portions of the content were assisted by LLMs. Specifically, the LLMs were used to optimize language, enhance the fluency of the literature review, and assist with organizing and refining the structure of certain sections. However, all final content was reviewed and revised by the authors, based on their own research and analysis, to ensure the accuracy and originality of the work. The use of the LLMs was intended to improve writing efficiency and did not influence the core ideas, data analysis, or conclusions of the paper.

## B    TRAINING DETAILS

### B.1    TRAINING HYPERPARAMETERS

Table 4 presents the hyperparameters used to fine-tune the Qwen2-1.5B models with MoSELoRA on two tasks: language understanding and mathematical reasoning. The same hyperparameter settings are applied to both tasks. Each experiment is conducted independently, with a single run for each model. The final model trained is used for evaluation. For the baseline methods, the same hyperparameter configuration is reused.

Table 4: Hyperparameters for MoSELoRA.

| Hyperparameter | Value |
|---|---|
| Base Model | Qwen2-1.5B |
| Num Train Epoch | 3 |
| Optimizer | AdamW |
| Weight Decay | 0.01 |
| Warmup Ratio | 0.1 |
| Learning Rate | $3 \times 10^{-4}$ |
| Target Modules | gate_proj, down_proj, up_proj |

### B.2    STATISTICS OF THE LANGUAGE UNDERSTANDING DATASET

We conduct experiments using a subset of the General Language Understanding Evaluation (GLUE) dataset(Wang et al., 2018), a benchmark for for training, evaluating, and analyzing natural language understanding systems. Specifically, we selected datasets Microsoft Research Paraphrase Corpus (MRPC)(Dolan & Brockett, 2005) and Corpus of Linguistic Acceptability (CoLA)(Warstadt et al., 2019). As shown in Table 5, this dataset consists of consist of various training and testing examples, each designed to evaluate specific linguistic tasks,including semantic equivalence and grammatical acceptability.

Table 5: The detailed statistics of language understanding datasets.

| Dataset | Train | Test | Task Description |
|---|---|---|---|
| MRPC | 3,668 | 1,725 | Determine if a pair of sentences are semantically equivalent |
| CoLA | 8,551 | 1,043 | Evaluate the grammatical acceptability of English sentences |

### B.3    STATISTICS OF MATHEMATICAL REASONING DATASETS

As illustrated in the Table6, we have constructed a mathematical reasoning training set consisting of a mix of four datasets, totaling 13,262 examples. These datasets include GSM8K (Cobbe et al., 2021), MAWPS (Koncel-Kedziorski et al., 2016), SVAMP (Patel et al., 2021), and subset of AQuA(Ling et al., 2017), each focusing on different aspects of mathematical reasoning. Additionally, we have incorporated MATH500(Lightman et al., 2023) into the test set to further evaluate the model's performance.

## C    GRADIENT BOUND ANALYSIS

The following presents a gradient analysis of our MoSELoRA framework, demonstrating that the gradients in our design remain bounded.

**Bound on the gradient w.r.t.** $u$.    Let $u = \phi(z)$ and define the lifting coordinate

$$a_\kappa(u) = \sqrt{\mathrm{sgn}(-\kappa)\,\|u\|^2 + \varphi(\kappa)}, \qquad \varphi(\kappa) = \begin{cases} 1/|\kappa|, & \kappa \neq 0, \\ 0, & \kappa = 0\,. \end{cases}$$

Table 6: The detailed statistics of mathematical reasoning datasets.

| Dataset | Data Number | Task Type |
|---------|-------------|-----------|
| Train | 13,262 | Mixed |
| Test | | |
| GSM8K | 1,319 | Question-Answering |
| MAWPS | 520 | Question-Answering |
| SVAMP | 300 | Question-Answering |
| MATH500 | 500 | Question-Answering |
| AQuA | 254 | Option |

A direct differentiation gives

$$\nabla_u a_\kappa(u) = \frac{\text{sgn}(-\kappa)\,u}{\sqrt{\text{sgn}(-\kappa)\,\|u\|^2 + \varphi(\kappa)}}, \qquad \big\|\nabla_u a_\kappa(u)\big\| = \frac{\|u\|}{\sqrt{\text{sgn}(-\kappa)\,\|u\|^2 + \varphi(\kappa)}}. \qquad (11)$$

**Case $\kappa < 0$ (hyperbolic).** Here $\text{sgn}(-\kappa) = 1$ and $\varphi(\kappa) = 1/|\kappa|$. Hence

$$\big\|\nabla_u a_\kappa(u)\big\| = \frac{\|u\|}{\sqrt{\|u\|^2 + 1/|\kappa|}} \leq 1 \quad \text{for all } u,$$

so $a_\kappa$ is globally 1-Lipschitz in $u$.

**Case $\kappa > 0$ (spherical).** Now $\text{sgn}(-\kappa) = -1$ and the domain requires $\|u\|^2 \leq 1/|\kappa|$. Let $R = \sup \|u\| < 1/\sqrt{|\kappa|}$, which can be enforced by a bounded activation together with scaling/projection. Then, from equation 11,

$$\big\|\nabla_u a_\kappa(u)\big\| = \frac{\|u\|}{\sqrt{1/|\kappa| - \|u\|^2}} \leq \frac{R}{\sqrt{1/|\kappa| - R^2}} =: C_u(\kappa, R) < \infty.$$

Therefore, as long as $u$ stays at a fixed margin from the boundary (e.g., $R = (1-\varepsilon)/\sqrt{|\kappa|}$ with $\varepsilon > 0$), the gradient is uniformly bounded.