# OpenReview forum: "Parameter-Efficient Fine-Tuning of LLMs with Mixture of Space Experts"
_ICLR.cc/2026/Conference — Submitted to ICLR 2026_

### Official Review · Reviewer_tjm7 · 2025-10-29

**Soundness:** 3
**Presentation:** 3
**Contribution:** 3
**Rating:** 6
**Confidence:** 4

**Summary:**

In this paper, the authors propose an MoSELoRA framework for parameter-efficient fine-tuning (PEFT) of large language models (LLMs). This framework integrates multiple geometric spaces such as Euclidean, hyperbolic, and spherical, into a unified architecture. The key idea is that these different linguistic structures are better represented in different geometric manifolds. In this regard, the MoSELoRA extends Low-Rank Adaptation (LoRA) by incorporating a Mixture of Space (MoS) approach, where tokens are dynamically routed to geometric experts based on input context. The authors introduce a lightweight routing mechanism and curvature-aware optimization to reduce computational overhead and improve training stability. Experiments on natural language understanding and mathematical reasoning benchmarks demonstrate consistent improvements over strong baselines.

**Strengths:**

1. The integration of multiple constant-curvature spaces into PEFT is original and well-motivated by linguistic and structural properties of language.
2. The use of stereographic projection and Lorentz model for efficient manifold transitions is elegant and avoids expensive exp/log mappings.
3. The paper includes curvature dynamics analysis, optimizer comparisons, and single-space expert variants to validate design choices.
4. The proposed approach demonstrates speedups and reduced parameter activation compared to other MoE-based methods.
5. MoSELoRA outperforms state-of-the-art PEFT methods across multiple benchmarks, especially in mathematical reasoning tasks.

**Weaknesses:**

1. The paper is focused entirely on LLMs and NLP benchmarks. It is unclear how this approach will be applied to multimodal LLMs unless extended to vision-language models or geometric representation learning in vision.
2. The routing mechanism that assigns tokens to geometric experts is described as “lightweight” and based on token-level projections, but the mathematical formulation lacks clarity. There is no theoretical guarantee that the routing leads to optimal space selection.
3. The paper does not analyze the stability or convergence properties of curvature learning. For instance, how does curvature interact with gradient flow in high-dimensional manifolds? Also, does the routing converge to a stable assignment? Is it differentiable and robust to noise?
4. There is no in-depth analysis of why certain tasks (e.g., math reasoning) benefit from hyperbolic embeddings beyond empirical observation. A theoretical framework linking task structure to manifold geometry would strengthen the claims.
5. While the method generalizes well to unseen math problems, its performance on other domains (e.g., commonsense reasoning, multi-hop QA) is not explored.

**Questions:**

1. Can MoSELoRA be extended to vision-language models? Have the authors considered applications in geometric scene understanding? The current evaluation is limited to LLMs and NLP benchmarks, with no experiments on multi-modal reasoning or vision-language grounding.
2. Is there any theoretical guarantee that the routing leads to optimal space selection? Could the authors provide visualizations of token embeddings across different spaces to support interpretability?
3. Does the routing converge to a stable assignment, and is it robust to noise? Are curvature parameters sensitive to initialization, and would curvature regularization improve stability?
4. Can the authors provide a theoretical framework linking task structure to manifold geometry?
5. The authors do not provide the source code. Can they clarify implementation details or release the code to ensure reproducibility?

---

> ### Author Response · Authors · 2025-12-03
>
> **Q1**: Extension beyond NLP and applicability to vision-language models.
>
> **A1**: We note that the MoSELoRA design is not inherently restricted to language-only tasks. The core mechanism—selective routing across experts defined on different geometric manifolds—is broadly applicable to multi-modal reasoning, and we anticipate semantic advantages in geometric scene understanding as well. In this work, we follow prior baselines and evaluate on established NLP benchmarks to isolate textual generalization effects, but we view extending MoSELoRA to vision-language grounding and geometric perception as a promising direction for future work.
>
> ---
> **Q2**: Interpretability of routing and evidence of space selection.
>
> **A2**: We understand the reviewer’s interest in interpretability. Since both curvature and expert routing in MoSELoRA are trainable parameters, the model is free to adaptively select the most appropriate space for each token and its contextual semantics during training. While we do not assume predefined geometric suitability for individual tokens, we observe that the model actively explores transitions between geometries and converges toward stable curvature configurations in later stages of training. Visualizing token embeddings and routing dynamics is indeed valuable, and we plan to incorporate such analyses in extended research iterations.
>
> ---
> **Q3**: The routing mechanism that assigns tokens to geometric experts is described as “lightweight” and based on token-level projections, but the mathematical formulation lacks clarity.
>
> **A3**: Existing methods heavily depend on exponential and logarithmic mappings, requiring repeated use of sinh, cosh, and inverse hyperbolic functions, like
> $\sinh(\cosh^{-1}(\kappa\langle \mathbf{x},\mathbf{y}\rangle_{\mathcal{L}}))$, which are prohibitively expensive and inherently difficult to parallelize in practice. By comparison, our method eliminates such computational burdens, relying solely on inner-product and square-root operations. This design leads to substantial efficiency and memory gains, which are clearly reflected in our experimental results and supported in our reply to Reviewer uMja.
>
> ---
> **Q4**: Routing stability and sensitivity of curvature initialization.
>
> **A4**: We provide theoretical analysis (please refer to Appendix C) to ensure that the effects of curvature initialization on gradient norms remain bounded by a constant factor, preventing instability due to curvature-induced distortions. In practice, we find the routing assignments to become more stable over training and resilient to noise perturbations. We agree that curvature regularization could further improve stability, and exploring such constraints is a compelling direction for further study.
>
> ---
> **Q5**: Theoretical linkage between task structure and manifold geometry.
>
> **A5**: We acknowledge the reviewer’s interest in theoretical grounding. Empirically, linguistic token distributions often exhibit scale-free patterns where token frequency decreases exponentially with token norm; this aligns well with the exponential expansion properties of hyperbolic space. However, this relationship does not universally hold across all datasets or all tokenizers, motivating the need to incorporate multiple geometric spaces—including Euclidean space as a special-case manifold—to better accommodate diverse structural priors present in different tasks. This analysis can be found in **Section 5.3** and **Figure 3** in the revised version. We observe that the token frequency distribution exhibits a scale-free behavior, reflecting the hyperbolic statistical characteristics of the underlying training corpus. Meanwhile, the token-norm distribution shows an exponential growth pattern on the right tail, whereas the left side deviates from such behavior. Prior work has partially observed similar phenomena, but their analysis focuses primarily on hyperbolic spaces, which are particularly well-suited for modeling power-law structures—yet less capable of capturing the broader range of non-Euclidean geometries that emerge across different contexts and domains.
>
> ---
> **Q6**: Performance on other domains (e.g., commonsense reasoning, multi-hop QA) is not explored.
>
> **A6**: Beside the Math reasoning and natural language understanding, we add experiments related to QA and commonsense reasoning, which also illustrates the improved performance of our method. Detailed information can refer to Answers to Reviewer uMja.
>
> ---
> **Q7**: Implementation transparency and code availability.
>
> **A7**: We intend to release the code following publication to support reproducibility. Our implementation is based on modifications to the PEFT framework, with changes localized primarily to the adapter implementation in the tuner component. The formulas, architectural behavior, and hyperparameter choices described in the paper are sufficient to reproduce the method with minimal additional effort.

---

### Official Review · Reviewer_MtEJ · 2025-10-29

**Soundness:** 3
**Presentation:** 2
**Contribution:** 3
**Rating:** 4
**Confidence:** 3

**Summary:**

This paper introduces MoSELoRA, a new fine-tuning method that generalizes the Mixture of LoRA Experts (linear mapping in the Euclidean space) to Mixture of Space Experts (non-Euclidean geometric mappings, including hyperbolic and spherical spaces). The authors proposed a unified representation of the mapping for all three considered spaces. Numerical experiments have been reported on several standard fine-tuning benchmarks, demonstrating the effectiveness.

**Strengths:**

1. The idea of the mixture of spaces is interesting.
2. The development of the lightweight token routing mechanism and the unified mapping for three spaces is interesting.
3. The proposed simplification achieves an acceleration of the computation for the geometric mapping.

**Weaknesses:**

1. The overall scope of the experiments is somewhat limited in terms of the evaluated models, model sizes, datasets, and tasks. Expanding the experimental coverage would strengthen the empirical claims.
2. The paper remains somewhat vague in several important aspects. The clarity and presentation could be improved by providing more details, such as:
- how each expert is selected during the forward pass
- how the routing value is computed for each token and expert
- what the auxiliary loss is
- what is the resulting full fine-tuning method
- which rank is used for each method in the Table in the experiments
- what are the hyperparameters that are used for all other baseline methods and are they optimally tuned,
- what is the total number of trainable parameters for all the other baseline methods.

3. The advantage of the proposed method over using only the hyperbolic space appears relatively small for the current setup. It might be more interesting to see the following experiments:

 - The paper notes that different geometric spaces may be better suited to different datasets. Suppose dataset A aligns best with Euclidean geometry. Would the trained MoSELoRA model, after being trained on a large and diverse dataset (as in the current setup), tend to select the Euclidean space more frequently during inference when being evaluated on dataset A?

- Conversely, if MoSELoRA were trained only on dataset A (which is well-suited to the Euclidean space), would it again favor the Euclidean space more often during training and inference?

**Questions:**

1. It is a bit surprising that LoRA with a higher rank and DoRA underperform the base LoRA.
2. Are there any reasons why the target modules do not include q,k, and v projections?

---

> ### Author Response · Authors · 2025-12-03
>
> **Q1**: The overall scope of the experiments is somewhat limited in terms of the evaluated models, model sizes, datasets, and tasks.
>
> **A1**: To address the concerns about scalability with respect to model architecture and size, we have additionally applied our method to Qwen2.5-3B, enabling evaluation on a larger model with different structural characteristics. We also add three new benchmarks, including commonsenseQA(CSQA) and openbookQA(OBQA). The results below demonstrate the performance of MoSELoRA compared to baseline methods under a consistent hyperparameter search setting.
>
> | Methods (Base: Qwen2.5-3B) | Param (%) | GSM8k | MATH500 | MAWPS | SVAMP | AQuA | MRPC | CoLA | RTE | CSQA | OBQA | Avg |
> |---|---:|---:|---:|---:|---:|---:|---:|---:|---:|---:|---:|---:|
> | Base              | 0      | 59.06 | 0.40 | 60.19 | 74.67 | 50.00 | 56.41 | 62.80 | 51.62 | 32.40 | 32.40 | 47.99 |
> | LoRA (r=32)       | 1.441  | 65.50 | 25.60 | 65.58 | 76.33 | 38.98 | 87.71 | 69.13 | 89.53 | 82.80 | 87.00 | 68.81 |
> | HydraLoRA (r=8)   | 2.036  | 64.52 | 22.40 | 65.77 | 75.33 | **42.91** | 88.46 | 85.91 | 89.89 | 82.56 | 89.20 | 70.69 |
> | HypLoRA (r=32)    | 1.441  | 64.67 | 26.40 | 65.77 | **78.67** | 41.34 | 88.23 | **87.34** | 90.25 | 79.85 | 87.80 | 71.03 |
> | **Ours (r=8)**    | 2.036  | **63.68** | **29.60** | **81.73** | **78.00** | **39.37** | **88.41** | **86.19** | **90.97** | **82.47** | **89.0** | **72.94** |
>
> ---
> **Q2**: The clarity and presentation could be improved by providing more details, like expert selecting, routing value, auxiliary loss, lora rank, hyper-parameters and activated param.
>
> **A2**: Each token is passed through a routing module that predicts routing scores for all experts. Similar to standard MoE architectures, the router then selects the top-k experts for each token based on these routing scores. The routing values are computed via a learned routing network applied to the token representation, producing per-expert scores. These scores are then normalized and used to select the top-k experts for execution.
>
> For our MoS-LoRA, we evaluate both rank-8 and rank-64 settings. All baseline LoRA methods use rank-8 to ensure comparable trainable parameter budgets across methods.
>
> We keep batch size, number of epochs (=3), and maximum response length fixed across all methods. For learning rate, we perform a consistent sweep over the same search range (2e-5 to 4e-3). For baselines, we additionally tune the LoRA scaling factor via alpha/rank ∈ {2, 4}. For each method, we report results from the best-performing configuration under this shared hyperparameter search.
>
> Since baselines use rank-8 and rank-32 for all experiments, the number of trainable parameters is kept approximately consistent with our rank-8 configuration, allowing a fair comparison. We have listed the specific values in the "Param" column of the table.

---

> > ### Author Response · Authors · 2025-12-03
> >
> > **Q3**: It might be more interesting to see the following experiments.
> >
> > **A3**: We sincerely appreciate your insightful suggestion regarding improving interpretability by analyzing the downstream geometric suitability of the data. Indeed, examining the underlying distribution of the data is an important direction, and prior works have explored how to better estimate the latent geometric structure within datasets [1,2]. However, natural language tokens, due to their highly diverse contexts, often involve entirely different semantic relationships and distributional patterns simultaneously. This makes it difficult to pre-determine a single “most suitable” geometric space through pre-computation alone. In contrast, routing tokens to space experts operating in different geometric spaces at the same time for each token can naturally address this issue.
> >
> > To support our claim, we analyze and visualize common datasets for mathematical reasoning and commonsense reasoning using both Qwen2 and Llama-3.2. Prior work has shown that mathematical data could exhibit scale-free characteristics in token norms and frequency distributions, suggesting that hyperbolic space should enhance modeling performance. However, we find that such properties do not consistently hold in the latest models. Instead, our observations reveal that token distributions learned by modern LLMs are more complex, with the left and right tails demonstrating approximately exponential and linear growth, respectively. Empirically, after fine-tuning on our mixed mathematical dataset, we indeed observe that the curvatures of different experts tend to move toward hyperbolic space. However, the final performance is still inferior to using a mixture of experts across all three geometric spaces. This result further supports the existence of these complex latent patterns. We observe similar behavior in both commonsense reasoning and natural language understanding tasks.
> >
> > ---
> > **Q4**: It is a bit surprising that LoRA with a higher rank and DoRA underperform the base LoRA.
> >
> > **A4**: We appreciate the observation. The underperformance of higher-rank LoRA and DoRA compared to base LoRA in our low-resource setting is actually consistent with established empirical trends in the PEFT literature. As noted in prior work, increasing the LoRA rank in large models (e.g., Qwen2-1.5B) with limited downstream data often leads to overfitting due to excess adaptation capacity. DoRA, which applies multiplicative magnitude scaling on directional updates, can further amplify optimization noise under such conditions, resulting in slightly degraded performance.
> >
> > Importantly, we carefully validated our implementations against official codebases and reproduced the baseline results reported in the original LoRA and DoRA papers under comparable settings. The observed trends are therefore not due to implementation errors, but reflect genuine behavior in our experiment regimes.
> >
> > **Q5**: Are there any reasons why the target modules do not include q,k, and v projections?
> >
> > **A5**: We intentionally did not include q/k/v projections as target modules for two practical reasons. Modifying attention projections significantly increases computational and memory cost, especially when combined with our curvature-aware transformations. Our goal is to design a PEFT method that remains lightweight and efficient. What‘s more, many prior works (e.g., LoRA [3], AdaLoRA [4]) show that adapting the feed-forward blocks already provides most of the performance gains, while modifying q/k/v often yields marginal improvement but introduces additional instability. For these reasons, we follow the standard and widely adopted practice of applying PEFT only to FFN layers. Nevertheless, extending MoSELoRA to attention projections is an interesting direction for future work.
> >
> >
> > [1] Leland McInnes, John Healy, and James Melville. Umap: Uniform manifold approximation
> > and projection for dimension reduction. arXiv:1802.03426, 2018
> >
> > [2] Joshua B. Tenenbaum, Vin de Silva, and John C. Langford. A global geometric framework for
> > nonlinear dimensionality reduction. Science, 290(5500):2319–2323, 2000.
> >
> > [3] Hu E J, Shen Y, Wallis P, et al. Lora: Low-rank adaptation of large language models[J]. ICLR, 2022, 1(2): 3.
> >
> > [4] Zhang Q, Chen M, Bukharin A, et al. Adalora: Adaptive budget allocation for parameter-efficient fine-tuning[J]. arXiv preprint arXiv:2303.10512, 2023.

---

### Official Review · Reviewer_vqM8 · 2025-10-30

**Soundness:** 2
**Presentation:** 2
**Contribution:** 2
**Rating:** 2
**Confidence:** 4

**Summary:**

This paper presents a novel and compelling approach to Parameter-Efficient Fine-Tuning (PEFT) by integrating Low-Rank Adaptation (LoRA) with a Mixture-of-Experts (MoE) framework across heterogeneous geometric spaces (Euclidean, Hyperbolic, Spherical). The proposed MoS and MoSELoRA methods dynamically route tokens to the most suitable space, enhancing the model's ability to capture diverse semantic structures.

**Strengths:**

1) Mixture of Space (MoS) considers more kinds of constant curvature spaces with learnable Gaussian curvature.

2) Ablation experiments make sense.

3) A comprehensive introduction to the references is given.

**Weaknesses:**

Quality:
1) The proposed method has a serious flaw in its geometric interpretation, manifested as structural inversion. Specifically, linear layer transformations should be performed in the embedding space, and vector additions should be carried out in the full space to present a clear geometric meaning, not the other way around. Further, no experiment verifies the advantages of learnable curvature families over fixed curvature families, and no theoretical evidence indicates any advantage of introducing non-Euclidean geometric for LoRA fine-tuning. Besides, there is a lack of experimental comparison with HypLoRA [1] and HELM [2].

Clarity:
1) The fine structure of Figure 1 is unclear (projection misleading, no reflection of learnable curvature, no evident basis for routing selection).
2) The actual method associated with Figure 3 is unclear and there is no comparison with traditional MoE methods.
3) The preliminary knowledge is poorly organized. In fact, the discussion about the exponential map is not beneficial for understanding the proposed method in this paper.

Significance:
1) The performance improvement shown in Table 1 is not very significant.

[1] Hyperbolic fine tuning for large language models. arXiv preprint arXiv:2410.04010, 2024.

[2] Hyperbolic large language models via mixture-of-curvature experts. arXiv preprint arXiv:2505.24722, 2025.

**Questions:**

see the Weaknesses.

---

> ### Author Response · Authors · 2025-12-03
>
> **Q1**: The proposed method has a serious flaw in its geometric interpretation, manifested as structural inversion. Further, no experiment verifies the advantages of learnable curvature families over fixed curvature families, and no theoretical evidence indicates any advantage of introducing non-Euclidean geometric for LoRA fine-tuning. Comparison with HypLoRA [1] and HELM [2].
>
> **A1.1:** We thank the reviewer for raising the concern regarding the geometric interpretation.
> However, the claim of “structural inversion” results from a misunderstanding of our mapping pipeline. MoSELoRA does NOT apply linear operators on curved manifolds.
>
> (1) Stereographic inverse projection is a coordinate chart, not a manifold operation. The mapping $ρ^{-1}_κ(x_i) = (ξ_i; s_i^T)^T \ with\ s_i ∈ R^n$ is the standard Lorentz-model stereographic chart for constant-curvature manifolds. This produces an Euclidean coordinate representation of a manifold point, analogous to projecting a sphere onto a 2D map.
>
> (2) All linear layers act exclusively on the Euclidean chart. In Section 4 Eq. (6), the LoRA update operates only on the space-like component: $W s_i  (\ s_i ∈ R^n)$, not on the manifold embedding $(ξ_i, s_i^T)$. Thus the method strictly follows the canonical Riemannian-NN pipeline: chart → linear → projection, which is identical to the former works like hyperbolic neural nets [1], hyperbolic transformers [2].
>
> (3) The manifold geometry is preserved through curvature-dependent projection. The mapping $ρ_κ((ξ_i; s_i'^T)^T)$ restores the geometry of each constant-curvature space. Although the linear computation occurs in the coordinate chart, the resulting embeddings retain Hyperbolic/Spherical/Euclidean geometric structure due to curvature-aware projection.
>
> Therefore, MoSELoRA does NOT invert the geometric structure. It employs the mathematically standard coordinate-based formulation of Riemannian representation learning, without applying linear operations on curved manifolds.
>
> **A1.2**: Our MoSELoRA scheme primarily leverages the advantages of combining different curvature spaces, building upon this foundation to allow for automatic curvature adjustment and automatic space adaptation. Experimental results demonstrate that the design incorporating three distinct spaces inherently provides benefits, and enabling curvature adjustment further enhances performance.
>
> **A1.3, A1.4**: For theoretical evidence, please refer to geometric evidence in response to Reviewer tjm7. Regarding the comparisons with other geometric approaches: **HypLoRA introduces hyperbolic space and performs LoRA-style fine-tuning entirely within that space.** To address this point, we have added more experiments, including comparisons across different base models, to demonstrate the necessity of combining experts with different curvatures and from different geometric spaces, as well as the advantages of our MoSELoRA framework.
>
> | Methods         | Param (%) | GSM8k  | MATH500 | MAWPS  | SVAMP  | AQuA   | MRPC  | CoLA  | Avg    |
> |----------------|----------:|------:|--------:|-------:|-------:|-------:|------:|------:|-------:|
> | MoSELoRA-fixed | 2.541     | 45.41 | 17.4    | 73.27  | 64.33  | **32.28** | **88.46** | 86.67 | 58.26 |
> | **MoSELoRA**   | 2.541     | **47.23** | **17.8** | **75.96** | **64.67** | 30.71 | 88.23 | **87.63** | **58.89** |
>
>
> **HELM is a method that improves the pretrained model itself**, which is fundamentally different from our finetuning approach. The only relevant aspect is that HELM also uses experts with different hyperbolic curvatures. To highlight the necessity of combining experts from different geometric spaces, we include a baseline in our experiments that incorporates this idea; please refer to the *ablation study on hyperbolic experts* in the experiment section.
>
> ---
> **Q2**: The fine structure of Figure 1 is unclear (projection misleading, no reflection of learnable curvature, no evident basis for routing selection).
>
> **A2**: Thank you for your comments regarding the demonstration of our paper. We have revised the Figure 1 to make the following structures clear and comprehensive: space projection from the input space to all three spaces, learnable curvature in expert design, and our grouped load balance for isolated space experts. And we also want to clarify that the projection is an unified  mapping with respect to the curvature.
>
> [1] Ganea O, Bécigneul G, Hofmann T. Hyperbolic neural networks[J]. Advances in neural information processing systems, 2018, 31.
>
> [2] Menglin Yang, Harshit Verma, Delvin Ce Zhang, Jiahong Liu, Irwin King, and Rex Ying. 2024. Hypformer: Exploring Efficient Transformer Fully in Hyperbolic Space. In Proceedings of the 30th ACM SIGKDD Conference on Knowledge Discovery and Data Mining (KDD '24). Association for Computing Machinery, New York, NY, USA, 3770–3781. https://doi.org/10.1145/3637528.3672039

---

> > ### Author Response · Authors · 2025-12-03
> >
> > **Q3**: The actual method associated with Figure 3 is unclear and there is no comparison with traditional MoE methods.
> >
> > **A3**: Thank you for your question. Regarding Figure 3, we aim to demonstrate the efficiency advantage of our MoS scheme's high-performance space mapping approach over the commonly used exp/log mapping (which is illustrated in the Preliminary Section). Traditional methods, such as those exp/log mappings employed in HypLoRA, encounter efficiency bottlenecks when applied to LLM-scale models. To address this, we propose a scheme that significantly reduces such computational overhead. Comparative experiments show that our approach substantially decreases computation time and reduces GPU memory usage (please refer to answers to Reviewer **uMja** Q2).
> >
> > As for traditional MoE-based schemes, we would like to clarify that our method is designed for post-training PEFT fine-tuning and does not involve pre-training. Under this premise, we have compared our approach with several MoE or MoE-inspired PEFT schemes, such as Hydralora and MElora, to highlight the performance of our solution. Our method outperforms these schemes across multiple benchmark scores.
> >
> > ---
> > **Q4**: The preliminary knowledge is poorly organized. In fact, the discussion about the exponential map is not beneficial for understanding the proposed method in this paper.
> >
> > **A4**: Thank you for your comments regarding the organization of our paper. We would like to clarify the purpose of the Preliminary section. Its primary logical aim is to provide the necessary background, focusing on the foundational framework of our method—namely, the expert-based LoRA post-training fine-tuning scheme, which is also where our main innovations and improvements lie. Additionally, since non-Euclidean space research primarily relies on exponential and logarithmic maps for space transformations, and we propose a mapping scheme that achieves higher performance with greater efficiency and reduced memory usage, we have included the principles of the exp/log mapping scheme in this section as foundational content to facilitate comparison with our proposed approach.
> >
> > ---
> > **Q5**: The performance improvement shown in Table 1 is not very significant.
> >
> > **A5**: We wish to clarify that, compared to historical PEFT schemes, the performance improvement achieved by MoSELoRA is similar to that of existing schemes such as Hydralora and HMoRA over their respective baselines—typically a 2-3 percentage point gain on relevant benchmarks. It is worth noting, however, that our method achieves a **5% improvement on the highly challenging MATH500 reasoning dataset** and a **15.9% gain on MAWPS**. We believe that integrating our approach with popular reasoning models holds significant potential.

---

### Official Review · Reviewer_uMja · 2025-11-01

**Soundness:** 3
**Presentation:** 3
**Contribution:** 3
**Rating:** 4
**Confidence:** 4

**Summary:**

The paper introduces MoSELoRA (Mixture of Space Experts LoRA), a parameter-efficient fine-tuning (PEFT) framework that integrates heterogeneous constant-curvature spaces (hyperbolic, spherical, and Euclidean) within a unified Mixture of Space (MoS) formulation. Unlike prior LoRA variants that operate purely in Euclidean geometry, MoSELoRA dynamically assigns token representations to curvature-specific subspaces through a lightweight routing mechanism, thereby adapting the fine-tuning process to the intrinsic geometry of language data. The method learns curvature parameters end-to-end, stabilizes training via separate optimizers for curvature and LoRA weights, and ensures gradient boundedness across all geometric spaces. Empirical results demonstrate consistent improvements on benchmarks.

**Strengths:**

1. The paper presents a well-motivated observation that existing PEFT methods assume flat Euclidean geometry, which may be suboptimal for hierarchical or cyclic semantic structures. MoSELoRA bridges this gap by modeling curvature diversity through a mixture of geometric experts.
2. The lightweight routing and space-mapping method achieves over 4x speedup over standard exp–log scheme
3. The curvature evolution analysis demonstrates interpretable geometry adaptation (Fig. 2): lower transformer layers remain near-Euclidean, while higher layers evolve toward hyperbolic or spherical curvature depending on task semantics

**Weaknesses:**

1. Experiments are limited to the Qwen2-1.5B model. There is no analysis on how this method scales for larger models or different architecture
2. Efficiency is only reported in terms of runtime, it is unclear if there is any memory overhead or advantages
3. The motivation for employing the curvature spaces (hyperbolic, spherical, Euclidean) is well supported by prior literature. However, the paper does not empirically justify the necessity of using all three simultaneously. Table 3 compares single-space variants against the full mixture but omits two-expert combinations (e.g., hyperbolic + Euclidean), leaving it unclear whether the full tri-expert setup is essential or if similar benefits could be achieved with fewer experts and reduced complexity.

**Questions:**

Please refer to the weaknesses.
* Would integrating MoSELoRA with quantization-based (e.g., QLoRA [1]) or gradient-efficient PEFT methods (e.g., GaLore [2]) yield similar performance and efficiency gains?

---

> ### Author Response · Authors · 2025-12-03
>
> Thanks for your valuable comments and suggestions.
>
> **Q1**: Experiments are limited to the Qwen2-1.5B model. There is no analysis on how this method scales for larger models or different architecture.
>
> **A1**: For the concerns about the scope of the model architecture and model sizes, besides qwen2-1.5B, we adapt our model to qwen2.5-3B to test our model on different architectures and model sizes, with three added datasets on commonsenseQA, openbookQA and natural language understanding. Here is the result comparing our MoSELoRA with the baselines using the same searching range of hyperparameters.
> | Methods (Base: Qwen2.5-3B) | Param (%) | GSM8k | MATH500 | MAWPS | SVAMP | AQuA | MRPC | CoLA | RTE | CSQA | OBQA | Avg |
> |---|---:|---:|---:|---:|---:|---:|---:|---:|---:|---:|---:|---:|
> | Base              | 0      | 59.06 | 0.40 | 60.19 | 74.67 | 50.00 | 56.41 | 62.80 | 51.62 | 32.40 | 32.40 | 47.99 |
> | LoRA (r=32)       | 1.441  | 65.50 | 25.60 | 65.58 | 76.33 | 38.98 | 87.71 | 69.13 | 89.53 | 82.80 | 87.00 | 68.81 |
> | HydraLoRA (r=8)   | 2.036  | 64.52 | 22.40 | 65.77 | 75.33 | **42.91** | 88.46 | 85.91 | 89.89 | 82.56 | 89.20 | 70.69 |
> | HypLoRA (r=32)    | 1.441  | 64.67 | 26.40 | 65.77 | **78.67** | 41.34 | 88.23 | **87.34** | 90.25 | 79.85 | 87.80 | 71.03 |
> | **Ours (r=8)**    | 2.036  | **63.68** | **29.60** | **81.73** | **78.00** | **39.37** | **88.41** | **86.19** | **90.97** | **82.47** | **89.0** | **72.94** |
>
>
> ---
> **Q2**: Efficiency is only reported in terms of runtime, it is unclear if there is any memory overhead or advantages
>
> **A2**: We understand the reviewer’s concern that our previous efficiency analysis focused primarily on runtime and did not clearly address potential memory overhead. In response, we have expanded our evaluation to separately measure both one forward and backward computation costs on average. Our MoSLoRA design introduces only moderate memory overhead:
> | Method   | Total Time (ms) | Phase         | Avg Time (ms) | Avg Memory (MB) |
> |----------|-----------------|---------------|---------------|-----------------|
> | MoS      | **57.456**      | forward_avg   | 2.163         | 277.241         |
> |          |                 | backward_avg  | 2.117         | 324.0           |
> | Exp/Log  | **135.075**     | forward_avg   | 2.626         | 565.642         |
> |          |                 | backward_avg  | 8.636         | 396.0           |
>
> - **Forward pass**: MoSLoRA uses **~277 MB**, compared to **~566 MB** for Exp/Log.
> - **Backward pass**: MoSLoRA requires **~324 MB**, which is comparable to the baselines.
>
> Overall, these results demonstrate that, in addition to runtime benefits, MoSLoRA maintains competitive or lower memory consumption, particularly in the forward pass, while achieving superior efficiency.
>
> ---
> **Q3**: The motivation for employing the curvature spaces (hyperbolic, spherical, Euclidean) is well supported by prior literature. However, the paper does not empirically justify the necessity of using all three simultaneously. Table 3 compares single-space variants against the full mixture but omits two-expert combinations (e.g., hyperbolic + Euclidean), leaving it unclear whether the full tri-expert setup is essential or if similar benefits could be achieved with fewer experts and reduced complexity.
>
> **A3**: We thank you for raising this insightful concern regarding the necessity of combining all three curvature spaces in our MoSELoRA. We fully understand the question about whether comparable benefits might be achieved with fewer experts and reduced complexity. Therefore, we have added new experiments evaluating two-expert combinations, specifically pairing Euclidean, hyperbolic, and spherical experts in all possible two-way configurations. The updated results are provided below.
> | Method                  | GSM8k | MATH500 | MAWPS  | SVAMP | AQuA  | MRPC  | CoLA | Avg    |
> |------------------------|------:|--------:|-------:|------:|------:|------:|-----:|-------:|
> | Mixture of EH experts  | 44.20 | 16.6    | 72.5   | 60.33 | 27.56 | 87.19 | 86.67 | 56.44 |
> | Mixture of HS experts  | 45.26 | 15.8    | 73.65  | 64.00 | 27.95 | 88.93 | 87.06 | 57.52 |
> | Mixture of ES experts  | 43.14 | 15.0    | 74.62  | 60.67 | 32.67 | 88.00 | 86.67 | 57.25 |
> | **MoSELoRA**           | 47.23 | 17.80   | 75.96  | 64.67 | 30.71 | 88.23 | 87.63 | **58.89** |

---

> > ### Author Response · Authors · 2025-12-03
> >
> > **Q4**: Would integrating MoSELoRA with quantization-based (e.g., QLoRA [1]) or gradient-efficient PEFT methods (e.g., GaLore [2]) yield similar performance and efficiency gains?
> >
> > **A4**: Thank you for the insightful question. MoSELoRA is designed to be orthogonal to both quantization-based methods (such as QLoRA) and gradient-efficient PEFT approaches (such as GaLore), since our adaptation operates only on the LoRA update path and does not modify the underlying forward or backward computation graph of the backbone transformer. In principle, integrating MoSELoRA with QLoRA or GaLore should be straightforward, as these methods focus on complementary aspects—memory compression and gradient efficiency—rather than geometric representation. However, our current work focuses on isolating the effects of geometry-aware adaptation itself, and we leave systematic integration and evaluation with these techniques as promising future extensions.
> >
> > We sincerely appreciate the reviewer’s insightful feedback, which has significantly helped us refine both the clarity and rigor of the manuscript.

---

### Meta-Review · Area_Chair_MhTk · 2026-01-13

**Summary:**

This paper proposes MoSELoRA, a geometry-aware parameter-efficient fine-tuning framework that extends LoRA to a very interesting idea - Mixture of Space (MoS): Euclidean, hyperbolic, and spherical experts with learnable curvature and token-level routing.
The submission is technically solid, empirically validated, and intellectually interesting. Although the reviewers ask for more models, more datasets, and more ablations, the rebuttal is pretty strong. Overall, MoSELoRA is an empirically useful new approach.

**Reviewer Concerns:**

Most reviewers concerns are addressed, reviewer vqM8's concern can not be addressed.

**Reviewer Scores:**

The reviewers uMja and MtEJ could increase their scores but not much

---

### Decision · Program_Chairs · 2026-01-26

Reject